# Comprehensive Metabolomic Profiling in Adults with X-Linked Hypophosphatemia: A Case-Control Study

**DOI:** 10.3390/biomedicines13010022

**Published:** 2024-12-26

**Authors:** Luis Carlos López-Romero, José Jesús Broseta, Marta Roca-Marugán, Noemí Máñez Ramírez, Julio Hernández-Jaras

**Affiliations:** 1Department of Nephrology, Hospital de Manises, 46940 Valencia, Spain; 2Department of Nephrology and Renal Transplantation, Hospital Clínic of Barcelona, 08036 Barcelona, Spain; jjbroseta@clinic.cat; 3Metabolomics Unit, Health Research Institute Hospital La Fe (IIS La Fe), 46026 Valencia, Spain; marta_roca@iislafe.es; 4Day Hospital Hematology and Oncology, Nursing Service, Hospital Universitari i Politècnic La Fe, 46026 Valencia, Spain; noemimanyez@gmail.com; 5Department of Nephrology, Hospital Universitari i Politècnic La Fe, 46026 Valencia, Spain; hernandez_jul@gva.es

**Keywords:** X-linked hypophosphatemia, XLH, metabolites, metabolomics, tubulopathy, hypophosphatemia

## Abstract

Background: X-linked hypophosphatemia (XLH) is a rare disorder characterized by elevated levels of fibroblast growth factor 23 (FGF-23), leading to hypophosphatemia and complications in diagnosis due to its clinical heterogeneity. Metabolomic analysis, which examines metabolites as the final products of cellular processes, is a powerful tool for identifying in vivo biochemical changes, serving as biomarkers of pathological abnormalities, and revealing previously uncharted metabolic pathways. Methods: A multicenter cross-sectional case-control study of adult patients diagnosed with XLH was conducted. Serum metabolomic analysis was performed with an Ultra-Performance Liquid Chromatography equipment (UPLC) coupled to a high-resolution mass spectrometer (MS). An analysis of metabolic pathways using MetaboAnalyst version 5.0 and a quantitative enrichment analysis (QEA) was performed. We employed multivariate statistical models, including a principal component analysis (PCA) and an orthogonal partial least squares discriminant analysis (OPLS-DA) regression model. Results: A cohort of 20 XLH patients and 19 control subjects were recruited. A total of 104 metabolites were identified. The differential metabolites identified included glycine, taurine, hypotaurine, phosphoethanolamine, pyruvate, guanidoacetic acid, serine, succinate, 2-aminobutyric acid, glutamine, 2-hydroxyvaleric acid, methionine, ornithine, phosphorylcholine, hypoxanthine, lysine, and N-methylnicotinamide. Enrichment analysis identified disturbances in key metabolic pathways, including phosphatidylethanolamine biosynthesis, sphingolipid metabolism, and phosphatidylcholine biosynthesis. Additionally, pathways related to cysteine metabolism, glycolysis, and pyruvate metabolism. Conclusions: This study identified significant differences in the metabolic profiles of individuals with XLH compared to healthy controls. These findings enhance understanding of potential pathogenic mechanisms and offer a metabolic basis for further in-depth investigations into XLH.

## 1. Introduction

The renal phosphate wasting syndromes represent a heterogeneous group of diseases with various etiologies, all biochemically manifesting as hypophosphatemia. X-linked hypophosphatemia (XLH), classified as a rare disease according to European criteria, is the most prevalent inherited form of rickets, with an estimated prevalence of approximately 4.8 per 100,000 [1,2]. This condition arises from loss-of-function mutations in the gene encoding the X-linked phosphate regulatory endopeptidase homolog (PHEX). These mutations lead to excess circulating fibroblast growth factor 23 (FGF-23), the primary regulator of renal phosphate homeostasis. Increased FGF-23 reduces renal phosphate reabsorption, leading to hypophosphatemia and decreased 1,25-dihydroxyvitamin D synthesis [3].

Clinical manifestations such as osteomalacia or rickets, short stature, malformations in the lower limbs, muscle or joint pain, fractures, or pseudofractures characterize the classic phenotype of XLH subjects. Most cases are diagnosed during childhood, but phenotypic expression can vary. There are cases of classic non-phenotypic XLH patients with muscle pain and weakness that may be the only clinical manifestation. In such cases, clinical suspicion is crucial for requesting diagnostic tests to avoid late diagnosis. The variability in presentation is diverse even among individuals of the same family, and there has been no established phenotype-genotype relationship [4].

Metabolites are the end products of cellular regulatory and metabolic processes. They provide specific and unique information about a cell’s physiological state, offering insight into its phenotype. In contrast to genes or proteins, metabolites are often easily accessible [5]. There is an increasing amount of the literature discussing the use of metabolites to influence processes, including stem cell differentiation, cell signaling, and immune responses. This suggests the potential for identifying and utilizing metabolites to impact phenotypes, a concept that is both intriguing and promising in the field of genetics and metabolomics [6,7].

Metabolomics is utilized to identify the array of metabolites associated with physiological conditions or aberrant processes. Comparing different metabolomic profiles can yield new insights into the biochemical mechanisms underlying disease pathophysiology, facilitate biomarker discovery, and identify metabolites capable of altering a cells or an organism’s phenotype [8].

Metabolomics is classified into untargeted and targeted approaches. Untargeted metabolomics provides unbiased analysis of a broad range of metabolites, including unknown ones, offering insights into global metabolic profiles and their links to phenotypic traits. Targeted metabolomics, by contrast, focuses on specific pathways, quantifying known metabolites with greater sensitivity and precision. Together, these approaches complement discovery and validation in metabolic research [9].

The application of liquid chromatography–high resolution mass spectrometry (LC-HRMS) in clinical practice has facilitated the identification of novel biomarkers and the development of targeted therapies, particularly in fields such as oncology [10]. Moreover, LC-HRMS has been employed in therapeutic drug monitoring, enabling enhanced precision in individual treatment, thereby improving therapeutic efficacy and minimizing adverse effects [11,12]. Collectively, these advancements highlight the pivotal role of LC-HRMS in advancing individualized patient care, fostering innovations in less invasive diagnostic approaches, and driving the development of novel therapeutic strategies.

The broad spectrum of clinical presentations and the complex nature of diagnosis associated with XLH mandate the pursuit of advanced methodologies that can address these intricate challenges. Consequently, we have opted to conduct a metabolomic investigation of individuals diagnosed with XLH, juxtaposed with a control cohort, to gain comprehensive insights into metabolite alterations. This study aims to lead to the generation of hypotheses regarding the pathophysiological mechanisms of the disease and future lines of research.

## 2. Materials and Methods

### 2.1. Study Design and Population

Adults with a clinical diagnosis of XLH were eligible to participate in this multicenter, cross-sectional, case-control study. Among these individuals, age- and gender-matched subjects were included at a ratio of approximately 1:1 for a total of 20 XLH patients and 19 subjects without the disease. Its main aim was to evaluate the metabolomic profile of XLH patients in comparison with a healthy control group. To carry out the study and ensure the maximum number of participants, a systematic cohort identification search using electronic health records was used. A detailed description of this tool is provided in our previous publication [13]. All participants provided informed consent prior to any study procedures being performed. The study was designed, conducted, recorded, and reported according to the guidelines of the Declaration of Helsinki data protection regulations and approved by the institutional review board and ethics committee of La Fe Health Research Institute (protocol code 2021-016-1, date of approval 27 January 2021).

In the case group, inclusion criteria included age greater than or equal to 18 years, a clinical diagnosis of X-linked hypophosphatemia documented by qualified clinicians, and meeting one or more of the following criteria: a documented PHEX mutation or variant of unknown significance in the patient or a family member with X-linked dominant inheritance, presence of at least one family member with a diagnosis of X-linked hypophosphatemia, or a documented serum intact FGF23 level greater than 60 pg/mL. Patients with hypophosphatemia secondary to non-renal loss or to other genetic etiologies, tumor-induced osteomalacia, and other renal tubulopathies were excluded from the study.

### 2.2. Data Collection and Study Assessments

Demographic, clinical, disease-related complications, radiological, and laboratory data were collected retrospectively. Subjects were scheduled for a single medical visit at the research unit within the Nephrology Service of Hospital Universitari i Politècnic La Fe to confirm medical record data. The clinical variables studied included the presence or absence of rickets or osteomalacia, lower limb deformities (leg length discrepancy, genu varum, genu valgum, tibial curvature, abnormal gait), muscle or bone pain, oral and dental disease, and fractures or pseudofractures.

Blood pressure (BP) was measured following standard protocols and by trained nurses or physicians using validated devices. Body weight and height were measured and used to estimate BMI, calculated as weight in kilograms divided by height in square meters. Blood and urine samples were collected from all subjects between 7 a.m. and 9 a.m. after following at least 8 h of fasting before the assessments. Intact FGF23 levels were determined using an enzyme-linked immunosorbent assay (ELISA) test kit from Kainos Laboratories, Tokyo, Japan. The estimated glomerular filtration rate (eGFR) was calculated using the Chronic Kidney Disease Epidemiology Collaboration (CKD-EPI) equation, while the remaining biochemical variables were assessed in accordance with the established protocols and best practice guidelines by Universitari I Politècnic La Fe Hospital. In cases where previous imaging was not available, a renal ultrasound was performed to assess the presence or absence of nephrocalcinosis or renal lithiasis. Available skeletal radiographs or computed tomography were reviewed by a radiologist. The treatment regimens of all participants were recorded.

Blood chemistry evaluations included phosphate, calcium, intact parathyroid hormone, 1,25-dihydroxyvitamin D, alkaline phosphatase, hemoglobin, creatinine, uric acid, total cholesterol, low-density lipoprotein cholesterol (LDL-C), high-density lipoprotein cholesterol (HDL-C), and triglycerides. Urinalysis evaluated phosphate, calcium, albumin and creatinine levels. The ratio of tubular transport maximum reabsorption of phosphate to glomerular filtration rate (TmP/GFR) and urine calcium/creatinine ratio was calculated.

*Hypertension* was defined as systolic BP ≥ 130 mm Hg and/or diastolic BP ≥ 80 mm Hg according to the 2017 ACC/AHA guidelines [14] and subjects on antihypertensive treatment or having the diagnosis documented in their medical records. *Chronic kidney disease* was defined as a GFR of less than 60 mL/min or the presence of structural damage according to KDIGO guidelines [15]. Nephrocalcinosis was defined as diffused calcium deposition within the kidney, identifiable through ultrasonography.

*Osteomalacia* was determined based on the definition of osteomalacia stated in the *Diagnosis Manual of Rickets and Osteomalacia* [16]. Diagnosis of *hyperparathyroidism* (HPT) was confirmed in patients with persistently elevated intact parathormone (PTH) levels greater than 65 pg/mL over a duration of six months or more.

### 2.3. Metabolic Profiling

#### Sample Collection and Preparation

For the metabolomic study, blood samples were collected in tubes containing ethylene diamine tetra acetic acid (EDTA) and processed within 30 min to prevent platelet activation and the breakdown of thermolabile compounds. Samples were promptly centrifuged at room temperature using an Eppendorf Centrifuge 5702 model (Eppendorf AG, based in Hamburg, Germany). at 5000 rpm for 10 min. The resultant serum was aliquoted into 250 μL portions and stored at −80 °C in a New Brunswick Scientific Ultra-Low Temperature Freezer V 410 Premium (Eppendorf AG, based in Hamburg, Germany) to ensure optimal preservation for further analysis.

All case and control samples were collected and stored to perform the analysis simultaneously. After thawing the samples, 20 µL of serum was mixed with 180 µL of cold methanol to separate protein-bound metabolites. The mixture was shaken at −20 °C for 30 min. Following this, the samples were centrifuged at 16,000 rpm for 20 min at 4 °C to separate the cellular fraction from the plasma. From each sample, 90 µL of the clear supernatant was carefully collected and transferred to an HPLC vial. Finally, 10 µL of a standard solution containing internal standards (a mixture of 20 µM caffeine-d9, leucine enkephalin, reserpine, and phenylalanine-d5) was added to the vial.

Quality control (QC) samples were prepared by pooling 10 μL from each extract. Blank samples, made by replacing the extract with ultrapure water and following the same sample preparation process as the real plasma ones, were used to identify potential artifacts and analyzed at the end of the sequence. An initial analysis of at least five QC injections were conducted at the start of the sequence to condition the column and equipment. These data were excluded from the multivariate analysis. Subsequently, a QC sample was analyzed for every 8 samples. The samples and QC were injected into the chromatographic system in a randomized order to ensure the analysis’s quality and reproducibility and reduce variation within the batch [17].

### 2.4. Untargeted Metabolomic Analysis

The metabolomic analysis was carried out in the Analytical Unit of the Medical Research Institute Hospital La Fe by an Ultra-Performance Liquid Chromatography equipment (UPLC) coupled to a high-resolution mass spectrometer (MS) with Orbitrap detector, UPLC-Q-Exactive Plus (Thermo Fisher, Waltham, Massachusetts, USA.). Chromatographic separation utilized a Hydrophilic interaction chromatography (HILIC), UPLC XBridge BEH amide column (150 mm × 2.1 mm, particle size 2.5 µm, Waters, Wexford, Ireland). The column was kept at 25 °C, while the auto-sampler was maintained at 4 °C, with an injection volume of 5 µL. The chromatogram ran for a total of 25 min at a flow rate of 105 µL/min. Elution of metabolites was achieved using gradients of mobile phase A (H_2_O and 10 mM ammonium acetate) and mobile phase B (acetonitrile). The elution gradient was: 90% mobile phase B and 10% mobile phase A for the first 2 min, 85% B from 2 to 3 min, 75% B from 3 to 8 min, 70% B from 8 to 10 min, 50% B from 10 to 13 min, 25% B from 13 to 16 min, 0% B from 16 to 22 min, and back to 90% B from 22 to 25 min.

The mass spectrometer was operated in full scan mode with an MS1 scan range of m/z 70–700 and 700–1700 and a resolving power of 140,000. Other MS parameters were as follows: sheath gas flow rate, 28 (arbitrary units); aux gas flow rate, 10 (arbitrary units); sweep gas flow rate, 1 (arbitrary units); spray voltage, 3.3 kV; capillary temperature, 320 °C; S-lens RF level, 65; AGC target, 3E6 and maximum injection time, 200 ms. Data-dependent fragmentation mode (DDA) was also employed for fragmentation purposes, and MS2 spectra were collected at 25eV energy with other instrument settings at resolution 17,500, AGC target 106, Maximum IT 250 ms, and isolation window 1.5 *m*/*z*. All data were acquired in centroid mode. All reagents and chemicals were purchased from Sigma Aldrich (St. Louis, MO, USA) and Fisher Scientific (ACN).

#### Metabolomic Data Processing and Potential Metabolites Annotation

The raw data obtained from the analysis were initially converted to mzXML format using the Mass Converter Proteowizard program. These files were then processed using the EI-MAVEN software (V2.10) for several tasks: alignment, noise filtering, integration of chromatographic peaks, generation of a peak table containing *m*/*z* retention times, and intensities of polar compounds. Peak areas were extracted and annotated using an in-house polar compound library. Data from positive and negative modes were merged for further analysis. If several metabolites had been detected in both ionization modes, we were left with the most abundant and best peak shape. Before the statistical analysis, data quality (reproducibility, stability) was evaluated using the internal standard’s stability and the QC’s coefficients of variation (CVs). Those molecular features with CVs > 30% were removed from the data matrix, and the filtered peak table was used for statistical analysis.

For data processing in EI-MAVEN, the configuration parameters were set as follows: ionization in automatic mode type ESI and precision q1 and q3 for MRN transitions at 0.5 amu with a total filter line. The maximum differences in retention time between spikes in a group were 15 s. The smoothness threshold was 2 λ, and the asymmetry was 0.08 ρ. A weighted average of the m/z values observed in the chromatographic peak was calculated to annotate the chromatographic peaks. For peak picking, the minimum intensity was set at 1000 with a maximum signal-to-noise ratio of 3 and a minimum peak width of 5 scans. Each peak was assigned a clustering score. The predefined tolerance for matching was ±10ppm. Peaks were classified as metabolites or artifacts based on their match with local database formulas. Adducts, fragments, or iso-topes of metabolites were considered artifacts. Background peaks were removed by comparing them with blank samples; peaks were eliminated if their intensity was greater than 0.5 times that of the sample.

### 2.5. Metabolic Pathways Analysis

We performed an analysis of metabolic pathways using MetaboAnalyst version 5.0. to reduce systematic bias and enhance consistency. This analysis allows the identification of biologically significant patterns that are enriched in the quantitative metabolomic data. The pathway analysis module combines results from pathway enrichment analysis with pathway topology analysis to identify the most relevant pathways involved.

Enrichment analysis and metabolite heatmap visualization were performed using MetaboAnalyst. We employed quantitative enrichment analysis (QEA), which requires a compound concentration table to be subsequently compared with compounds contained in the metabolite set library. A generalized linear model was used to estimate a Q-statistic for each metabolite set, which is the average of the Q-statistics for each metabolite in the set. The enrichment analysis was based on GlobalTest, which supports enrichment analysis with binary, multi-group, and continuous phenotypes. Since our focus is to identify the most relevant pathways within the pathway library, we are more interested in the pathway rank rather than its absolute *p*-value.

For pathway topology analysis, we used node centrality measures to estimate node importance, specifically betweenness centrality, which measures the number of shortest paths passing through a node. Here, we considered only the out-degree for the node importance measure, assuming that upstream nodes have regulatory roles over downstream nodes, and not vice versa. Given that we were testing multiple pathways simultaneously, the statistical *p*-values from the enrichment analysis were further adjusted for multiple testing. We described the total number of compounds in the pathway; hits or the number of actual matches from the uploaded data; raw *p* or the original *p*-value calculated from the enrichment analysis; Holm *p* or the *p*-value adjusted by the Holm–Bonferroni method; FDR *p* or the *p*-value adjusted using the false discovery rate; and the pathway impact value calculated from the pathway topology analysis.

### 2.6. Statistical Analysis

Qualitative data are described as frequencies, and quantitative variables are presented as mean and standard deviation when normally distributed, or median and interquartile ranges (IQRs) when non-normally distributed, according to the Shapiro–Wilk test. Groups were compared with a *t*-test, chi-squared test, or Mann–Whitney *U*-test, as appropriate. the results were considered statistically significant with two-tailed analyses, *p* < 0.05.

Metabolite concentration peaks were evaluated in XLH subjects and compared with those in the control group. The analysis, performed using an in-house R script (Version 4.3.3), involved a univariate statistical approach. A t-test was conducted to assess differences between the case and control groups, and the findings were illustrated using a volcano plot. The methodological framework included a fold change (FC) metric and relied on the paired Student’s t-test for normally distributed data. When the data deviated from normality, as confirmed by the Shapiro–Wilks test, the Wilcoxon signed-rank test was applied. Following a false discovery rate adjustment using the Benjamini–Hochberg procedure, we selected molecular features that exhibited a significant combination of fold change (|log_2_ FC| > 2) and statistical significance (*p*-value < 0.05). These characteristics, showing a strong interplay of fold change and significance, were identified to minimize the potential for false positives.

A multivariate analysis was then carried out with the significant features selected in this previous analysis, by using the OMICS skin of the SIMCA software v.17 (Sartorious Stedim Biotech, Aubagne, France). To extract maximum information and identify behavioral patterns, an exploratory unsupervised principal component analysis (PCA) was conducted. This multivariate pattern recognition technique to discriminate cases from controls where each axis value explained the percentage of total variance contributed by each principal component is a process that involves examining the distribution of data to simplify variability and groups them into principal components. In the second step, a supervised orthogonal projection to latent structures discriminant analysis (OPLS-DA) was used to determine the main discriminant variables responsible for the differences between groups. The models’ validity and robustness were assessed using R^2^(Y) to measure goodness-of-fit and Q^2^(Y) to evaluate goodness-of-prediction, with a Q^2^(Y) value exceeding 0.5 set as the threshold for acceptable predictive ability.

For univariate analysis, statistical significance was defined as *p* < 0.05 in a two-tailed Student’s t-test. The variable importance in the projection (VIP) score estimates the importance of each feature in the model and is commonly used for feature selection. Metabolic compounds showing VIP score > 1 and a FC greater than 1.3 were considered statistically different between groups. For statistical analysis, IBM SPSS^®^ Statistics version 26 and graphics, with GraphPad version 9 were used.

## 3. Results

### 3.1. Characteristics of the Study Participants and Clinical and Biochemical Analyses

A cohort of 20 patients diagnosed with X-linked hypophosphatemia (XLH) and 19 age and gender-matched control subjects was recruited for this study. The median age of the participants was 43.5 years (IQR: 28.7–55.2 years). The median weight, height, and body mass index (BMI) were 71 kg (IQR: 51.5–80.5 kg), 166 cm (IQR: 154.5–175 cm), and 23.6 kg/m^2^ (IQR: 22–28 kg/m^2^), respectively. No statistically significant differences were observed between the XLH patients and control subjects regarding these anthropometric variables.

However, significant differences between the two groups were identified in several clinical parameters. XLH patients exhibited a higher prevalence of short stature, lower limb deformities, pain, osteoarticular symptoms, and hypertension compared to controls (*p* < 0.05 for all comparisons). In contrast, no significant differences were observed between the groups in terms of dental disease, hyperparathyroidism, depression or anxiety, fractures or pseudofractures, lithiasis, or nephrocalcinosis.

Biochemically, patients with XLH exhibited significantly higher levels of intact fibroblast growth factor 23 (iFGF-23) (211.96 ± 148.58 pg/mL vs. 43 ± 21.3 pg/mL; *p* < 0.001) and alkaline phosphatase (97.8 ± 43.6 U/L vs. 58.6 ± 12.8 U/L; *p* = 0.001), as well as reduced serum phosphorus levels (2.14 ± 0.5 mg/dL vs. 3.3 ± 0.5 mg/dl; *p* < 0.001) and tubular maximum reabsorption of phosphate per glomerular filtration rate (TmP/GFR) (2.12 ± 0.48 mg/dL vs. 3.3 ± 0.5 mg/dL; *p* < 0.001). No statistically significant differences were observed between the groups for creatinine, uric acid, total cholesterol, low-density lipoprotein (LDL), high-density lipoprotein (HDL), triglycerides, hemoglobin, calcium, 1,25-dihydroxyvitamin D [1,25(OH)_2_D], and intact parathyroid hormone (iPTH). These findings are summarized in Table 1.

### 3.2. Metabolomic Results

A total of 104 metabolites were identified across both groups, with detailed chemical properties provided in Appendix A. Statistically significant differences were observed in several metabolite classes, including amino acids, peptides and their analogues, sulfinic acids, keto acids, phosphate esters, fatty acids, and conjugates. The differential metabolites identified include glycine, taurine, hypotaurine, phosphoethanolamine, pyruvate, guanidoacetic acid, serine, succinate, 2-aminobutyric acid, glutamine, 2-hydroxyvaleric acid, methionine, ornithine, phosphorylcholine, hypoxanthine, lysine, and N-methylnicotinamide.

We employed both multivariate and univariate statistical analyses to identify metabolites with differential expression between groups. The identified differential metabolites were visualized using a volcano plot (Figure 1). To evaluate discrimination and prediction of metabolomic data, we employed multivariate statistical models, including a robust principal component analysis (PCA) and an orthogonal partial least squares discriminant analysis (OPLS-DA) regression model. The score plots from both the PCA (Figure 2A) and OPLS-DA (Figure 2B) demonstrated clear metabolomic differences between the two groups. Metabolic biomarkers were selected based on a *p*-value < 0.05 and a variable importance in projection (VIP) value > 1. The PCA, including the QC samples, has been provided in the Appendix A.

Box and whisker plots (Figure 3) were used to represent the relative intensities of the metabolites with statistical differences. All the described metabolites exhibited higher levels in XLH subjects compared to the control group, while pyruvate was lower. No correlation or weak correlation was presented in the correlation analysis between discriminative metabolites and levels iFGF23, 1,25(OH)2D, and iPTH among both groups (Pearson correlation coefficient > 0.7, *p* < 0.05) (Appendix A).

### 3.3. Pathway Enrichment Analysis Using MetaboAnalyst

A comprehensive metabolomics analysis was conducted to profile X-linked hypophosphatemia and identify key metabolites associated with the condition. Using MetaboAnalyst 5.0, pathway enrichment analysis was performed based on the Kyoto Encyclopedia of Genes and Genomes (KEGG) database. This approach enabled the identification of significant alterations in metabolic pathways associated with XLH. The results are presented graphically in Figure 4 and summarized in detail in Figure 5. Enrichment analysis identified the top 25 pathways involved in XLH subjects revealing significant disturbances in several key metabolic pathways, including phosphatidylethanolamine biosynthesis, sphingolipid metabolism, and phosphatidylcholine biosynthesis. Additionally, pathways related to cysteine metabolism, glycolysis, pyruvate metabolism, glucose-alanine cycle, gluconeogenesis, and pyrualdehyde degradation were notably affected. It is also important to highlight the disruption in phospholipid biosynthesis [18].

## 4. Discussion

Through non-targeted metabolomic analysis, this study revealed significant differences in the metabolic profiles of individuals with X-linked hypophosphatemia (XLH) compared to a healthy control group. Using a mass spectrometry-based metabolite profiling platform, we identified a panel of 17 metabolites associated with XLH status in a cohort of 20 XLH subjects. We further analyzed the metabolic pathways most involved with these differential metabolites.

The primary findings of our study include variations in phosphoethanolamine levels, along with alterations in the biosynthetic pathways of phosphatidylethanolamine and phosphatidylcholine, as well as sphingolipid metabolism. Phosphoethanolamines are essential for the synthesis of fundamental phospholipids, such as glycerophospholipids and sphingolipids, and play a crucial role in key enzymatic processes, including the synthesis of phosphatidylcholine and phosphatidylethanolamine [19]. These results have broader implications for understanding cellular membrane dynamics in XLH.

Moreover, phosphoethanolamine has clinical relevance as a biomarker for hypophosphatasia (HPP), a rare inherited metabolic bone disorder. In HPP, elevated urinary phosphoethanolamine levels reflect a deficiency in tissue-nonspecific alkaline phosphatase activity, disrupting bone mineralization due to altered inorganic pyrophosphate (PPi) metabolism [20]. However, despite phosphoethanolamine’s role in metabolic pathways relevant to bone health, studies did not reveal a relationship between phosphoethanolamine levels and bone mineral density or fracture risk [21]. This finding suggests that phosphoethanolamine may not be directly involved in bone density regulation or fracture susceptibility in XLH.

While alterations in membrane phospholipids have been extensively documented in essential hypertension, recent evidence has identified specific phospholipids—phosphatidylethanolamines (PE) and phosphatidylcholines (PC)—as potential predictors of incident hypertension [22]. The observed associations between circulating PE and PC levels and blood pressure suggest that phospholipid metabolites could play an important role in blood pressure regulation in XLH. Additionally, phosphatidylcholines and phosphatidylethanolamines have been associated with low bone mineral density (LBMD) and an increased risk of osteoporosis progression [23].

Phosphatidylcholines (PCs) and phosphatidylethanolamines (PEs) are the most abundant phospholipids within cell membranes, where they serve crucial functions in cellular energy metabolism, proliferation, adhesion, stress response, and apoptosis [19]. A landmark study involving a substantial cohort and extended follow-up period demonstrated that levels of various lipid species—particularly phosphatidylcholines, phosphatidylethanolamines, and ceramides—were significantly associated with coronary heart disease (CHD) risk [24]. These lipid biomarkers enhanced the predictive accuracy of CHD beyond that achieved with conventional risk factors alone in an American population. Similarly, another study identified elevated levels of six phosphatidylethanolamines as being linked to a higher incidence of stroke, underscoring the potential role of these lipid species as biomarkers for cardiovascular risk [25].

Similarly, the amino acid profiles of XLH subjects revealed significant differences compared to controls, with notable elevations in taurine, glycine, and serine levels. Taurine, while not a structural protein component, is a highly abundant, non-essential amino acid predominantly found in bone and muscle tissue. It is metabolically involved in numerous processes critical to bone development and calcium homeostasis [26,27]. Our study revealed significantly elevated levels of taurine and hypotaurine in XLH subjects; these findings are comparable to those observed in individuals with bone tumors, but not in conditions associated with decreased bone mineral density. Previous studies have highlighted an inverse relationship between taurine levels and osteoporosis, with metabolomic analyses showing reduced taurine concentrations in the blood of patients with osteoporosis and osteopenia compared to healthy controls [28]. This relationship has been further validated in premenopausal women and elderly individuals with low BMD and fractures [29,30]. These findings support taurine’s potential as a biomarker for bone loss, possibly linked to disruptions in the transsulfuration pathway caused by elevated plasma homocysteine (Hcy) levels [31].

Beyond its role in bone metabolism, taurine has been associated with bone tumors such as osteosarcoma and Ewing sarcoma in adolescents, as well as chondrosarcoma in adults. Metabolomic investigations have increasingly focused on identifying biomarkers for tumor diagnosis and prognosis. For instance, a non-targeted metabolomic analysis of serum from 65 osteosarcoma patients revealed elevated levels of ascorbic acid and taurine compared to healthy controls [32]. Furthermore, taurine levels decrease in response to osteosarcoma treatments, correlating with oxidative stress and reductions in several amino acids, suggesting a dynamic role for taurine in tumor biology and treatment response [33].

Glycine was another amino acid that showed significant differences between the two groups. Glycine plays an essential role in bone metabolism as a primary structural component of collagen. This finding aligns with previous studies that have associated glycine levels with bone mineral density (BMD), suggesting that increased bone resorption may elevate serum glycine levels, potentially explaining its inverse association with BMD. Hydroxyproline, another collagen residue commonly used as a marker of bone resorption, can be converted into glycine in the kidneys [34]. This conversion raises the possibility that glycine may also serve as an indirect marker of bone resorption, acting as a degradation product of hydroxyproline. Supporting this hypothesis, a small study reported higher plasma glycine levels in men with idiopathic osteoporosis compared to controls [35]. Additionally, other metabolomic studies have reported an inverse relationship between glycine levels and femoral neck BMD [36]. Glycine levels have been negatively correlated with bone density and structure while showing a positive association with cortical porosity, a factor linked to increased fracture risk. Notably, both serum and urinary glycine levels were significantly associated with all fractures, including major osteoporotic fractures [37].

Our study also identified significant differences in serine levels between XLH subjects and the control group. Serine, synthesized by glucose-dependent anabolic cells, plays a crucial role in protein synthesis. This metabolite has been implicated in the growth and repair of bone tissue through endochondral ossification, a process that relies on the proliferation and hypertrophy of chondrocytes [38]. Impaired chondrocyte function has been associated with delayed or absent fracture healing that characterizes XLH subjects. In addition, guanidooacetic acid (GAA), a derivative of glycine and the immediate metabolic precursor of creatine, plays a critical role in cellular bioenergetics involving high-energy phosphate compounds. While GAA deficiency has been reported in various neurological and muscular disorders [39], the elevated levels observed in XLH patients in our study are likely associated with increased glycine concentrations.

Consistent with the patterns observed for other amino acids, lysine levels were also elevated in XLH subjects. Lysine serves multiple functions, including roles in proteinogenesis and in the cross-linking of collagen polypeptides, which contributes to its high concentration in muscle tissue [40]. In one study, multiple linear regression analysis demonstrated a significant correlation between isoleucine levels, total cervical degeneration score, and collagen cross-linking [41]. Conversely, carboxymethyllysine (CML), an advanced glycation end-product, is present in collagen fibers and has been reported to accelerate osteoblastic cell apoptosis and alter osteocyte functioning CML levels have, in fact, been associated with an increased risk of hip fractures [42]. Additionally, a study revealed significant positive correlations between CML and serum levels of sclerostin and fibroblast growth factor 23 (FGF23), both of which are secreted by osteocytes and known to impair bone mineralization [43]. These findings suggest that CML may directly influence osteocytes within the bone matrix. 

In XLH subjects, we observed elevated levels of glutamine, which may be related to its effect on bones and joints. Glutamine, a highly abundant non-essential amino acid, plays a crucial role in bone homeostasis by serving as an alternative energy source and providing precursors for protein and nucleic acid synthesis [44]. Its metabolism regulates the bioenergetics of bone cells, including mesenchymal stem cells, osteoblasts, chondrocytes, and osteoclasts, influencing their proliferation, differentiation, and mineralization [45]. Disruptions in glutamine metabolism have been linked to conditions such as osteoporosis and osteoarthritis [46]. Notably, in osteoarthritis, glutamine treatment has demonstrated protective effects on chondrocytes, shielding them from heat stress and nitric oxide (NO)-induced apoptosis, which may contribute to slowing disease progression.

Similarly, elevated succinate levels were identified in XLH patients, potentially contributing to impaired bone healing and increased fracture risk. Succinate, a key intermediate in the citric acid cycle, functions as a signaling molecule involved in various cellular responses. In macrophages, succinate stimulates the production of interleukin-1β (IL-1β), a cytokine that plays a critical role in bone mineralization during the regeneration phase and promotes osteogenic differentiation [47]. However, prolonged IL-1β stimulation may activate inflammatory pathways, potentially disrupting bone repair processes, especially in the context of fractures. Furthermore, studies indicate that excessive IL-1β activity in the presence of elevated succinate can inhibit mineralization [48]. These findings suggest a complex interplay between succinate signaling and inflammatory responses in the pathophysiology of XLH-related skeletal abnormalities.

The role of gamma-aminobutyric acid (GABA) in musculoskeletal diseases remains poorly understood. GABA, the principal inhibitory neurotransmitter in the central nervous system, exerts its effects by binding to specific neuronal receptors, leading to membrane hyperpolarization and subsequent neuronal inhibition [49]. It is synthesized from glutamate via the enzyme L-glutamic acid decarboxylase, with pyridoxal phosphate serving as a cofactor. Notably, taurine and lysine have been shown to enhance both GABA synthesis and activity [50]. In our study, the elevated levels of these metabolites observed in XLH subjects could potentially explain the increased GABA levels detected in this cohort.

Similarly, elevated ornithine levels have been reported in individuals with osteoarthritis, including those with hip and knee involvement, compared to control subjects [51,52]. As a critical precursor for collagen and polyamine synthesis, increased ornithine levels may reflect a reparative response aimed at cartilage repair [53]. These findings highlight the potential role of metabolomic alterations in XLH and other musculoskeletal conditions.

Subjects with X-linked hypophosphatemia (XLH) exhibited elevated levels of methylnicotinamide (Me-NAM) compared to the control group. Me-NAM is an endogenous metabolite of nicotinamide, whose catabolism produces S-adenosylhomocysteine, a precursor to homocysteine, a molecule with established associations to cardiovascular disease [54]. In clinical research, elevated serum levels of methylnicotinamide have been linked to obesity, diabetes, and coronary heart disease, independently of traditional cardiovascular risk factors [55]. Additionally, serum Me-NAM levels have been inversely associated with left ventricular ejection fraction [56]. Despite these findings, it remains unclear whether increased Me-NAM levels represent a compensatory mechanism or serve as a biomarker for atherosclerosis. In a separate pediatric study, multiple metabolites, including methylnicotinamide and hypoxanthine, were identified as strongly correlated with arterial hypertension [57]. These findings suggest potential clinical implications, as patients with XLH are already recognized to be at a heightened risk for hypertension and, consequently, cardiovascular disease.

Pyruvate was the only metabolite found to be significantly lower in concentration among XLH patients compared to the control group. While its role in bone and musculoskeletal diseases remains poorly characterized, pyruvate is a key intermediate in cellular metabolism. In the Krebs cycle, pyruvate—derived from glycolysis, fatty acid oxidation, and amino acid catabolism—plays a critical role in generating adenosine triphosphate (ATP), reactive oxygen species (ROS), and small molecule metabolites essential for cellular energy production and calcium homeostasis [58]. Furthermore, a study identified lower pyruvate levels in hypertensive XLH patients compared to their normotensive counterparts [59]. However, the specific relationship between pyruvate levels and blood pressure regulation in this population warrants further investigation. On the other hand, we did not find relevant information regarding the role of 2-hydroxyvaleric acid or methionine in bone metabolism or cardiovascular diseases.

This study is not without limitations. First, its observational design prevents the establishment of causal relationships between the observed metabolic changes and XLH. Second, the relatively small sample size classifies this work as exploratory and hypothesis-generating, requiring validation in larger cohorts to strengthen its findings. Third, the study did not account for the effect of medications and their metabolism. Most participants were treated with active vitamin D and oral phosphorus supplements, yet no definitive conclusions regarding the impact of these treatments could be drawn. Lastly, the potential influence of gut flora on metabolic regulation was not explored in this study, representing another area for future investigation.

Nevertheless, the study possesses notable strengths. Despite the constraints of the sample size, the statistical power was sufficient to detect significant differences in metabolite profiles between groups. Our cohort of XLH patients is among the largest published in Europe and the most extensive involving adult XLH patients. Furthermore, this is the first metabolomics study on XLH subjects to include a control group. This is particularly significant given the rarity of this disease, such that each data point contributes critically to the body of knowledge. Additionally, the inclusion of unaffected controls provides a valuable comparative dimension, enhancing the study’s overall robustness. A key strength of this research lies in the application of metabolomics, a rapidly advancing “omics” science in medical research. Metabolomics offers transformative potential in uncovering disease mechanisms, particularly in rare disorders such as XLH. This field provides a foundation for personalized medicine, representing a pivotal advancement in the future of healthcare.

## Figures and Tables

**Figure 1 biomedicines-13-00022-f001:**
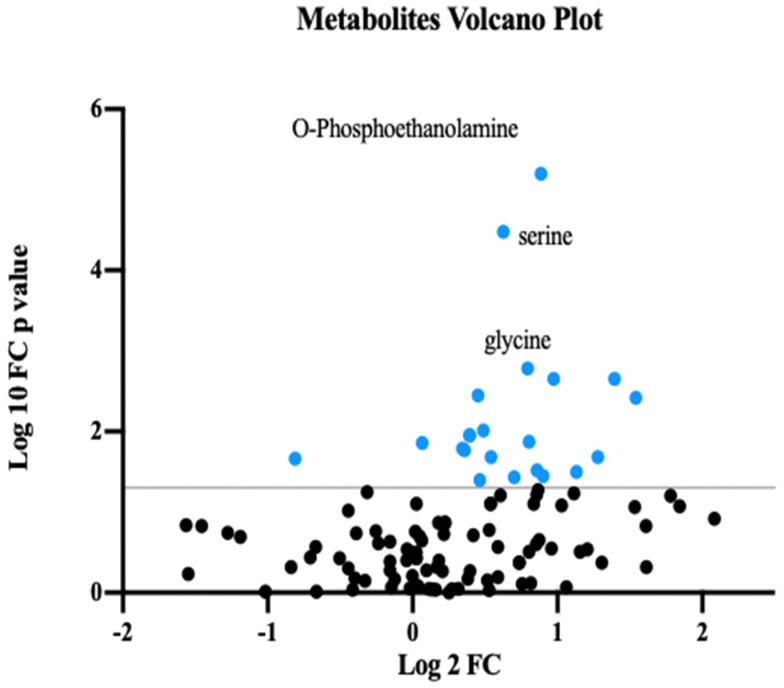
Volcano plot illustrating metabolomic differences between XLH patients and controls. Phosphoethanolamine, serine, and glycine emerged as the most discriminatory metabolites. The horizontal black line indicates the threshold for statistical significance, while blue dots highlight metabolites with significant alterations.

**Figure 2 biomedicines-13-00022-f002:**
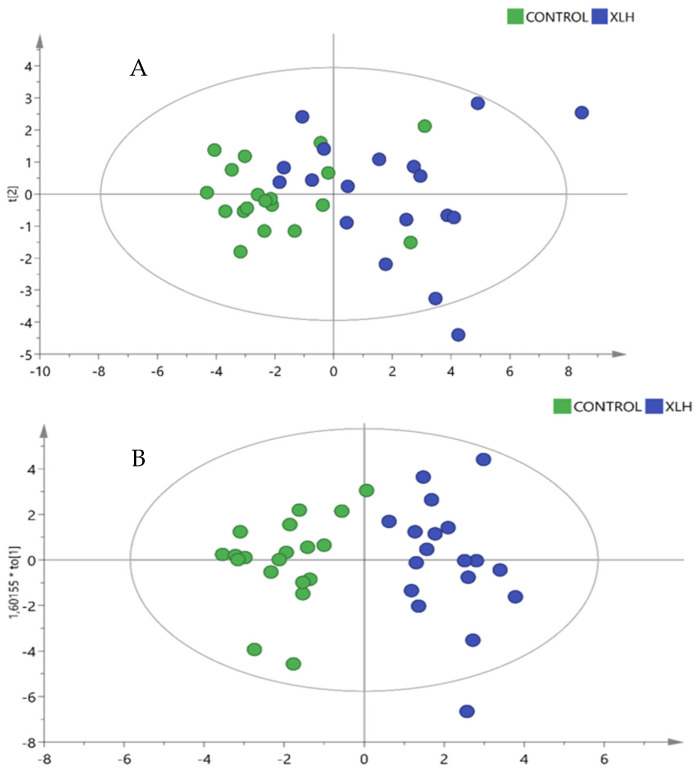
PCA score plot (**A**) and OPLS-DA score plot (**B**) illustrating the separation of metabolomic profiles between controls and XLH patients. Model validity and robustness were assessed using R^2^(Y) for goodness-of-fit and Q^2^(Y) for predictive accuracy, with Q^2^(Y) > 0.5 indicating acceptable predictive performance. R^2^ = 0.744, Q^2^ = 0.528.

**Figure 3 biomedicines-13-00022-f003:**
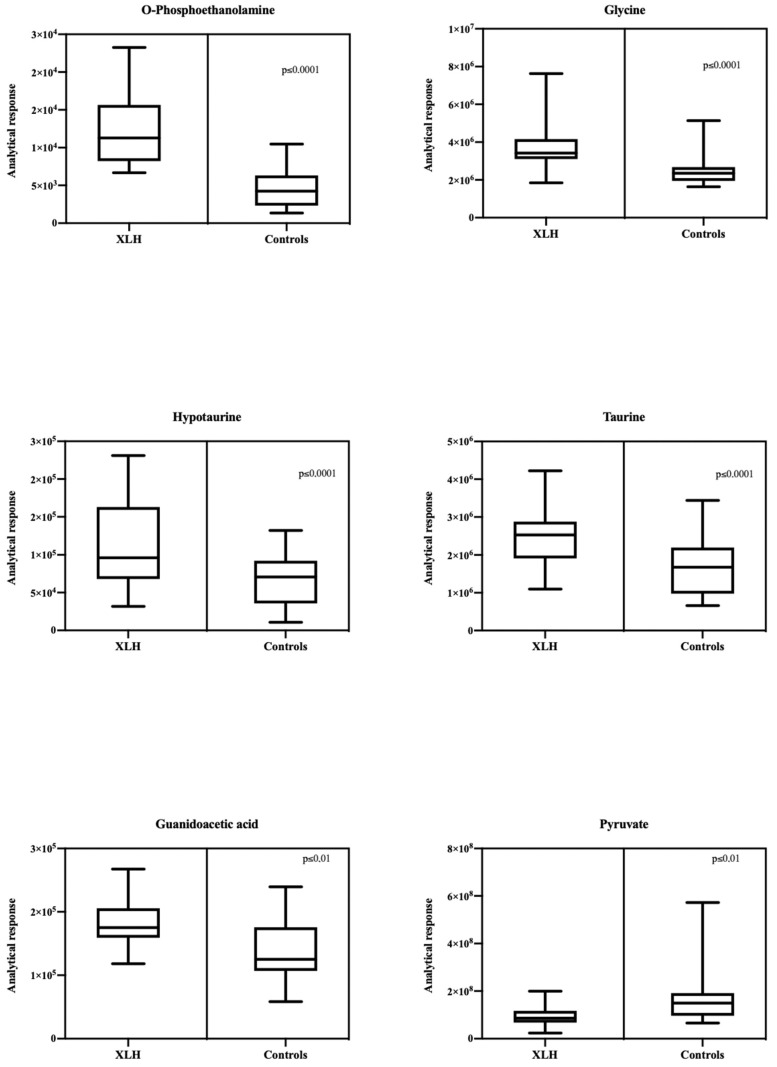
Box and whisker plots representing the relative intensities of the metabolites with statistical differences between XLH patients and controls.

**Figure 4 biomedicines-13-00022-f004:**
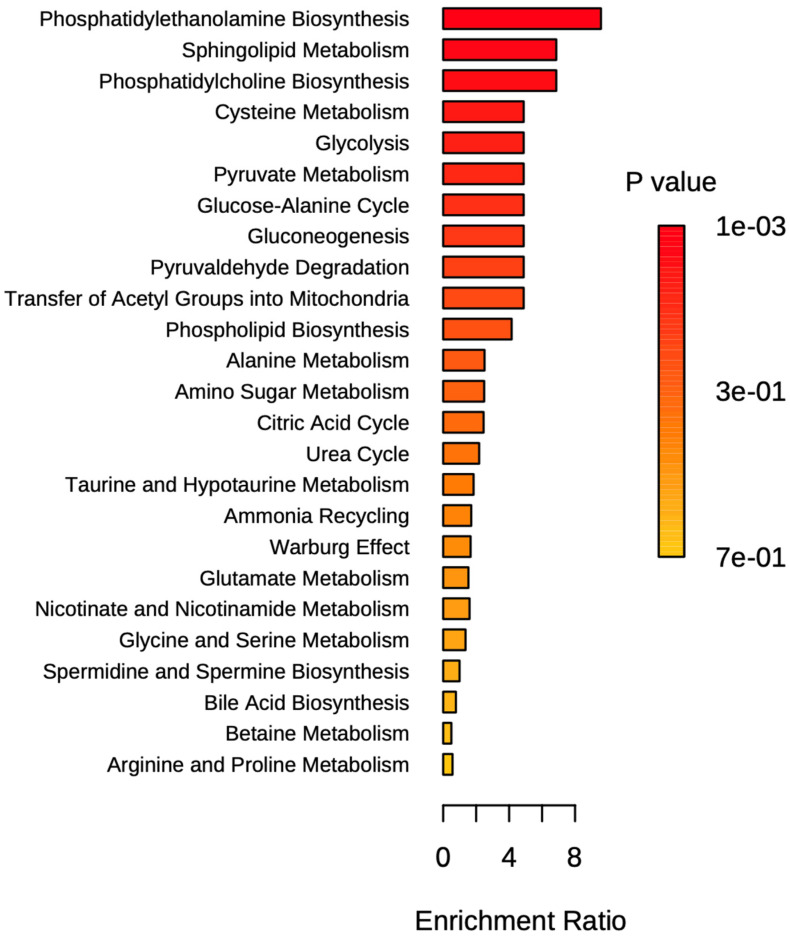
Pathway enrichment analysis of XLH patients reveals key metabolic pathways. The bar graph shows pathways ranked by their enrichment ratio, with darker colors indicating more statistically significant results (lower *p*-values). The x-axis represents the enrichment ratio, while the color scale denotes the *p*-value (1 × 10^−3^ to 7 × 10^−1^). Phosphatidylethanolamine biosynthesis, sphingolipid metabolism, and phosphatidylcholine biosynthesis exhibit the highest enrichment ratios and significance, suggesting their central role in the metabolic response.

**Figure 5 biomedicines-13-00022-f005:**
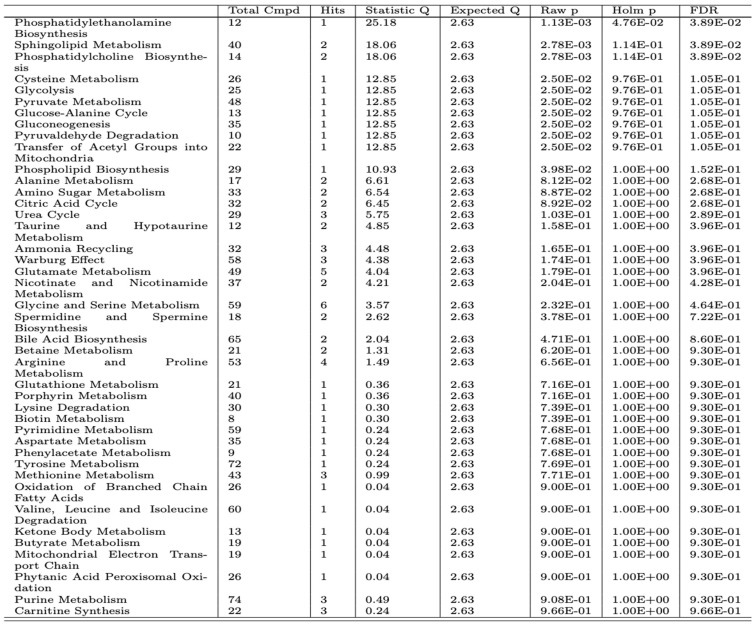
Quantitative enrichment analysis (QEA) performed using a package GloblaTest 3. It uses a generalized linear model to estimate a Q-statistic for each metabolite set. Columns include the pathway name and the total number of compounds in the pathway. Hits is the actually matched number from the data. Raw *p* is the original *p* value calculated from the enrichment analysis. Holm *p* is the *p* value adjusted by Holm–Bonferroni method. FDR *p* is the *p*-value adjusted using the false discovery rate.

**Table 1 biomedicines-13-00022-t001:** Demographic, clinical, and biochemical characteristics of the study population.

Variable	Total (*n* = 39)	XLH (*n* = 20)	Control (*n* = 19)	*p*-Value
Age, median (IQR)	43.5 (28.7–55.2))	44 (33.2–54.7)	35 (28–57.7)	NS
Male, n (%)	19 (48.7)	10 (50)	9 (47.3)	NS
Weight (kg), median (IQR)	71 (51.5–80.5)	73 (50.7–85.5)	69.5 (53.7–78.2)	NS
Height (cm)	166 (154.5–175)	158 (150–170.5)	169.5 (164–177)	0.02
BMI (Kg/m^2^)	23.6 (22–28)	27.5 (22–32.2)	23 (20.7–24)	0.01
Short stature, n (%)	13 (33.3)	12 (60)	1 (5.2)	0.000
Lower limb deformities, n (%)	9 (20.5)	9 (45)	0 (0)	0.000
Pain, n (%)	16 (41)	13 (65)	3 (15.7)	0.002
Osteoarticular symptoms, n (%)	17 (45.6)	15 (75)	2 (10)	0.000
Dental disease, n (%)	12 (30.8)	6 (30)	6 (31.6)	NS
Hypertension, n (%)	12 (30.7)	9 (45)	3 (15.8)	0.01
hyperparathyroidism, n (%)	9 (23)	7 (35)	2 (10.5)	NS
Depression or anxiety, n (%)	15 (38.4)	8 (40)	7 (36.8)	NS
Fractures/pseudofractures, n (%)	11 (28.9)	7 (35)	4 (21)	NS
Lithiasis, n (%)	3 (7.7)	1 (5)	2 (10.5)	NS
Nephrocalcinosis, n (%)	3 (7.7)	3 (15)	0 (0)	NS
Creatinine (mg/dL), mean ± SD	0.85 ± 0.2	0.91 ± 0.29	0.78 ± 0.09	NS
Uric acid (mg/dL), mean ± SD	5.4 ± 1.53	5.7 ± 1.52	5 ± 1.5	NS
Total cholesterol(mg/dL), mean ± SD	198.7 ± 37.4	197.1 ± 28.4	200.5 ± 46.4	NS
LDL-C (mg/dL), mean ± SD	118.6 ± 28.9	119 ± 24.3	118.2 ± 34.2	NS
HDL-C (mg/dL), mean ± SD	58.2 ± 15	55.3 ± 14.9	61.5 ± 15	NS
Triglycerides (mg/dL), mean ± SD	110.1 ± 52.7	115.9 ± 50.4	104 ± 55.9	NS
Hemoglobin (g/dL), mean ± SD	14.6 ± 1.5	14.6 ± 1.6	14.6 ± 1.4	NS
Alkaline phosphatase (UI/L), mean ± SD	79.3 ± 38	97.8 ± 43.6	58.6 ± 12.8	0.001
Phosphorus (mg/dL), mean ± SD	2.7 ± 0.8	2.14 ± 0.5	3.3 ± 0.5	0.000
Calcium (mg/dL), mean ± SD	9.4 ± 0.61	9.4 ± 0.7	9.4 ± 0.5	NS
Intact FGF23 (pg/mL), mean ± SD	127.3 ± 84.3	211.96 ± 148.58	43 ± 21.3	0.000
1,25(OH)_2_D (ng/mL), mean ± SD	27.2 ± 7.8	27.9 ± 9.4	26.3 ± 5.8	NS
iPTH (pg/mL), mean ± SD	63.8 ± 39.3	74.8 ± 48.9	51.6 ± 19.7	NS
TmP/GFR (mg/dL), mean ± SD	2.6 ± 0.8	2.12 ± 0.4	3.32 ± 0.5	0.000

BMI: body mass index, LDL-C: low-density lipoprotein cholesterol, HDL-C: high-density lipoprotein cholesterol, FGF23: fibroblast growing factor 23, Vit D: 1,25(OH)2 vitamin D, iPTH: intact parathormone, TmP/GFR: ratio of tubular maximum reabsorption of phosphate (TmP) to glomerular filtration rate (GFR), NS: Not significant.

## Data Availability

The data supporting the findings of this article are included in the main text of the manuscript. Additional data, such as all metabolomic results, are available upon request.

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
