# Peer review of "Comprehensive Metabolomic Profiling in Adults with X-Linked Hypophosphatemia: A Case-Control Study"

_biomedicines, 2024, doi:10.3390/biomedicines13010022_

Round 1

Reviewer 1 Report

Comments and Suggestions for Authors

In this paper, a non-targeted metabolomic study based on UPLC-HRMS was performed in human serum to investigate the mechanism of X-linked hypophosphatemia. The experimental design, the data collection, results and discussion appear well articulated and clearly presented. However, I have some questions and recommendations for the authors:

1. Line 81: What does “were included at a ratio of approximately 1:1” mean?

2. How was the randomization of the metabolomic sequences done?

3. How many blanks and QCs were injected? Were the samples injected in triplicate? This information should be reported in the main text.

4. Line 176: Is “85% B at 2 minutes” correct? Should it be “at 3 minutes” or “from 2 to 3 minutes”? If the conditions employed have been used in other papers, these should be referenced.

5. MS/MS conditions (such as fragmentation voltage, etc) should be included in section 2.

6. How was the data filtering carried out? What was the number of molecular features before and after the data filtering?

7. Metabolomic sequences were performed in both positive and negative modes. However, Figures 1 and 2 only show the results of one of the sequences without specifying the ionization mode. Can the authors provide the volcano plot, PCA, and OPLS-DA of both sequences?

8. The authors should provide the PCA including the QC samples.

9. Exact values of R2 and Q2 parameters should be mentioned in the main text or included in the figure.

10. I recommend that the authors use the MSI guidelines terminology to define metabolite identification (Sumner et al, Metabolomics 3, 2007, 211-221. DOI: 16.1007/s11306-007-0082-2).

11. Why was OPLS-DA used instead of PLS-DA?

12. Tables S1 and S2 should be mentioned in the main text.

13. Table S1 should include the main experimental MS/MS fragments of each metabolite.

Reviewer 2 Report

Comments and Suggestions for Authors

The current manuscript by Lopes-Romero et al aims to establish a new pipeline for the investigation of metabolomic profiling in adults with X-linked hypophosphatemiaveterinary by UPLC-HRMS. The topic interesting  and the results are looking promising. While the concept and design are very straightforward, the text suffers from many flaws as follows; 

1) Introduction section Lines 50-54

“....... Clinical manifestations such as osteomalacia or rickets, short stature, malformations  in the lower limbs, muscle or joint pain, fractures, or pseudofractures characterize the classic phenotype of XLH subjects. Most cases are diagnosed during childhood, but phenotypic expression can vary. There are cases of classic non-phenotypic XLH patients where muscle pain and weakness may be the only clinical manifestation.”...

 The authors published an article (ref 53) in similar concept and should explain what is difference when comparing with the current study. More details should be given.

2) Introduction section Lines 58-68

“.......Metabolites are the end products of cellular regulatory and metabolic processes. They  provide specific and unique information about a cell's physiological state, offering insight  into its phenotype. In contrast to genes or proteins, metabolites are often easily accessible  [5]. There is an increasing literature discussing the use of metabolites to influence pro-cesses, including stem cell differentiation, cell signaling, and immune responses. This suggests the potential for identifying and utilizing metabolites to impact phenotypes, a concept that is both intriguing and promising in the field of genetics and metabolomics [6-7].”.....

This part should include recent and relevant applications by LC-HRMS. In addition, more information should be given about screening strategies by LC-HRMS (e.g. target, non-target, retrospective)

3) Lines 138-164

This part needs refs.

4) Lines 165-234  

This part is really confusing.

What do you mean by “an Ultra-Performance Liquid Chromatography equipment (UPLC) coupled to a high-resolution mass spectrometer (MS) with Orbitrap UPLC-Q-Exactive Plus detector (UPLC-TOF/MS-Orbitrap, QExactive Plus MS)”, which I have never seen such system. The authors use UPLC-Q-Exactive MS or UPLC-Q-TOF-MS? Which one? Or this one as tandem is unique?

Another point is, ı wonder that the authors apply target or non-target methodology. How the authors achieve non-target analysis by LC-HRMS. It is not known how the authors constructed the databases constructed or what other databases were used. This is not clear. Also, methods are sparse, including missing details on the LC components of the system, column specifications, etc. Information on where analytical standards were purchased is missing, etc.

 5) line 337-362

This part needs refs and  It would be better to put MS spectra to be obtained, which improve article quality.

Round 2

Reviewer 1 Report

Comments and Suggestions for Authors

The comments and suggestions have been properly answered and added to the revised version of this manuscript. Therefore, I think it can be now published in Biomedicines

Reviewer 2 Report

Comments and Suggestions for Authors

The authors addressed all my concerns and the revised version is now suitable for publication.